METHODS AND RESOURCES

# Single-fly genome assemblies fill major phylogenomic gaps across the Drosophilidae Tree of Life

**Bernard Y. Kim**[1]*, **Hannah R. Gellert**[1], **Samuel H. Church**[2], **Anton Suvorov**[3], **Sean S. Anderson**[4], **Olga Barmina**[5], **Sofia G. Beskid**[1], **Aaron A. Comeault**[6], **K. Nicole Crown**[7], **Sarah E. Diamond**[7], **Steve Dorus**[8], **Takako Fujichika**[9], **James A. Hemker**[10], **Jan Hrcek**[11], **Maaria Kankare**[12], **Toru Katoh**[13], **Karl N. Magnacca**[14], **Ryan A. Martin**[7], **Teruyuki Matsunaga**[15], **Matthew J. Medeiros**[16], **Danny E. Miller**[17], **Scott Pitnick**[8], **Michele Schiffer**[18], **Sara Simoni**[1], **Tessa E. Steenwinkel**[19], **Zeeshan A. Syed**[8], **Aya Takahashi**[9], **Kevin H-C. Wei**[20], **Tsuya Yokoyama**[1], **Michael B. Eisen**[21,22], **Artyom Kopp**[5], **Daniel Matute**[4], **Darren J. Obbard**[23], **Patrick M. O'Grady**[24], **Donald K. Price**[25], **Masanori J. Toda**[26], **Thomas Werner**[27], **Dmitri A. Petrov**[1,28]*

**1** Department of Biology, Stanford University, Stanford, California, United States of America, **2** Department of Ecology and Evolutionary Biology, Yale University, New Haven, Connecticut United States of America, **3** Department of Biological Sciences, Virginia Tech, Blacksburg, Virginia, United States of America, **4** Department of Biology, University of North Carolina Chapel Hill, Chapel Hill, North Carolina, United States of America, **5** Department of Evolution and Ecology, University of California Davis, Davis, California, United States of America, **6** School of Environmental and Natural Sciences, Bangor University, Bangor, United Kingdom, **7** Department of Biology, Case Western Reserve University, Cleveland, Ohio, United States of America, **8** Center for Reproductive Evolution, Department of Biology, Syracuse University, Syracuse, New York, United States of America, **9** Department of Biological Sciences, Tokyo Metropolitan University, Tokyo, Japan, **10** Department of Developmental Biology, Stanford University, Stanford, California, United States of America, **11** Institute of Entomology, Biology Centre, Czech Academy of Sciences, České Budějovice, Czech Republic, **12** Department of Biological and Environmental Science, University of Jyväskylä, Jyväskylä, Finland, **13** Department of Biological Sciences, Hokkaido University, Sapporo, Japan, **14** Hawaii Invertebrate Program, Division of Forestry & Wildlife, Honolulu, Hawaii, United States of America, **15** Department of Complexity Science and Engineering, The University of Tokyo, Tokyo, Japan, **16** Pacific Biosciences Research Center, University of Hawai'i, Mānoa, Hawaii, United States of America, **17** Division of Genetic Medicine, Department of Pediatrics; Department of Laboratory Medicine and Pathology, University of Washington, Seattle, Washington, United States of America, **18** Daintree Rainforest Observatory, James Cook University, Townsville, Australia, **19** Baylor College of Medicine, Houston, Texas, United States of America, **20** Department of Zoology, The University of British Columbia, Vancouver, Canada, **21** Department of Cell and Molecular Biology, University of California Berkeley, Berkeley, California, United States of America, **22** Howard Hughes Medical Institute, University of California Berkeley, Berkeley, California, United States of America, **23** Institute of Ecology and Evolution, University of Edinburgh, Edinburgh, United Kingdom, **24** Department of Entomology, Cornell University, Ithaca, New York, United States of America, **25** School of Life Sciences, University of Nevada Las Vegas, Las Vegas, Nevada, United States of America, **26** Hokkaido University Museum, Hokkaido University, Sapporo, Japan, **27** Department of Biological Sciences, Michigan Technological University, Houghton, Michigan, United States of America, **28** CZ Biohub, Investigator, San Francisco, California, United States of America

\* bernardkim@stanford.edu (BYK); dpetrov@stanford.edu (DAP)

**Data Availability Statement:** All new data generated for this work are deposited at NCBI SRA and GenBank under BioProject PRJNA1020440. The whole-genome alignment is archived at Dryad

## Abstract

Long-read sequencing is driving rapid progress in genome assembly across all major groups of life, including species of the family Drosophilidae, a longtime model system for genetics, genomics, and evolution. We previously developed a cost-effective hybrid Oxford Nanopore (ONT) long-read and Illumina short-read sequencing approach and used it to assemble 101 drosophilid genomes from laboratory cultures, greatly increasing the number

(DOI: dx.doi.org/10.5061/dryad.x0k6djhrd). NCBI accessions and citations for public data used for this study are provided in S1 Table. Illumina-only assemblies generated from publicly available datasets (i.e., not generated by this work, S1 Table), RepeatModeler2 libraries, variant calls, diploid assemblies, genomes, and phylogenetic trees are archived at Zenodo (DOI: dx.doi.org/10.5281/zenodo.11200891). Raw Nanopore signal data (fast5, pod5) will be provided upon email request due to their large file sizes. Online locations of data are summarized in S8 Table.

**Funding:** SGB was funded by the National Science Foundation (NSF) GRFP. SHC was funded by NSF PRFB DBI 2109502. KNC was funded by the National Institutes of Health (NIH) R35 GM137834. SED and RAM were funded by The Expanding Horizons Initiative, Case Western Reserve University. SD, SD, and ZS were funded by the NSF DEB-1811805. MBE was funded by the Howard Hughes Medical Institute. TF was funded by the Grant-in-Aid for Japan Society for the Promotion of Science (JSPS) Research Fellow (22J11897). JAH was funded by the NIH NHGRI 5T32HG000044-27. JH was funded by the Czech Ministry of Education, Youth and Sports grant ERC CZ LL2001. MK was funded by the Academy of Finland 322980. BYK was funded by the NIH NIGMS F32 GM135998. AK was funded by the NIH NIGMS R35 GM122592. TM was funded by the Grant-in-Aid for Research Activity Start-up 22K20565. DM was funded by the NIH NIGMS R35GM148244. PMO was funded by NSF DEB2030129, DEB1839598, and DEB1241253. DJO was funded by the UK Biotechnology and Biological Sciences Research Council BB/T007516/1. DAP was funded by the NIH NIGMS R35GM118165. AT was funded by the JSPS KAKENHI JP23H02530. KHW was funded by the NIH NIGMS K99GM137041. TW was funded by the NSF DOB/DEB-1737877 and the Huron Mountain Wildlife Foundation (Michigan Tech Agreement #1802025). National Science Foundation: https://www.nsf.gov/; National Institutes of Health: https://www.nih.gov/; The Expanding Horizons Initiative, Case Western Reserve University: https://artsci.case.edu/expanding-horizons-initiative/; Howard Hughes Medical Institute: https://www.hhmi.org/; Japan Society for the Promotion of Science: https://www.jsps.go.jp/; Czech Ministry of Education, Youth and Sports: https://www.msmt.cz/; Academy of Finland: https://www.aka.fi/; UK Biotechnology and Biological Sciences Research Council: https://www.ukri.org/councils/bbsrc/; Huron Mountain Wildlife Foundation: https://www.hmwf.org/ The funders did not play a role in the study design, data

of genome assemblies for this taxonomic group. The next major challenge is to address the laboratory culture bias in taxon sampling by sequencing genomes of species that cannot easily be reared in the lab. Here, we build upon our previous methods to perform amplification-free ONT sequencing of single wild flies obtained either directly from the field or from ethanol-preserved specimens in museum collections, greatly improving the representation of lesser studied drosophilid taxa in whole-genome data. Using Illumina Novaseq X Plus and ONT P2 sequencers with R10.4.1 chemistry, we set a new benchmark for inexpensive hybrid genome assembly at US $150 per genome while assembling genomes from as little as 35 ng of genomic DNA from a single fly. We present 183 new genome assemblies for 179 species as a resource for drosophilid systematics, phylogenetics, and comparative genomics. Of these genomes, 62 are from pooled lab strains and 121 from single adult flies. Despite the sample limitations of working with small insects, most single-fly diploid assemblies are comparable in contiguity (>1 Mb contig N50), completeness (>98% complete dipteran BUSCOs), and accuracy (>QV40 genome-wide with ONT R10.4.1) to assemblies from inbred lines. We present a well-resolved multi-locus phylogeny for 360 drosophilid and 4 outgroup species encompassing all publicly available (as of August 2023) genomes for this group. Finally, we present a Progressive Cactus whole-genome, reference-free alignment built from a subset of 298 suitably high-quality drosophilid genomes. The new assemblies and alignment, along with updated laboratory protocols and computational pipelines, are released as an open resource and as a tool for studying evolution at the scale of an entire insect family.

## Introduction

Species in the model system Drosophilidae (vinegar or fruit flies) have long served to showcase the power of genomics as a tool for understanding evolutionary pattern and process. Drosophilid species were represented among the very first metazoan genomes [1,2], comparative genomic datasets [3,4]; population genomic datasets [5]; and more recently, single-cell atlases [6]. These community-built resources have served as a foundation for subsequent discoveries across many fields of scientific inquiry. A logical next step would be the exhaustive sequencing of the >4,400 species [7] in this biologically diverse family, with the ultimate aim of creating a framework for connecting micro- to macro-evolutionary processes through the powerful lens of the *Drosophila* system. This seemingly daunting goal is made entirely feasible by cost effective long-read sequencing approaches [8–10], which greatly simplify the process of genome assembly. Further, the genomic tractability of drosophilids (genome sizes approximately 140 to 500 Mbp), their worldwide abundance, and scientific importance of this model system, allows us to rapidly sequence new species while building upon the extensive scaffold of existing scientific resources and knowledge [11].

We previously assembled 101 genomes of 93 different drosophilid species in a major step towards the comprehensive genomic study of Drosophilidae [8]. A hybrid Oxford Nanopore (ONT) long-read and Illumina short-read assembly proved to be an inexpensive and efficient approach to this task, at the time enabling us to build genomes in under a week at the low cost of US $350 per genome. Since then, the number of drosophilid genomes has increased significantly: as of writing (August 2023), 468 genome assemblies for 167 drosophilid species are available to the public through NCBI (representative genomes are listed in **S1 Table**). While

collection and analysis, decision to publish, or preparation of the manuscript.

**Competing interests:** The authors have declared that no competing interests exist.

impressive, the genomic resources for the *Drosophila* system have major gaps: one of the most obvious is the sampling bias towards species predisposed to culture in a laboratory environment [7,11]. As a result, large sections of interesting drosophilid biodiversity are almost entirely unstudied with modern genomic tools. This includes the *Scaptomyza*-Hawaiian *Drosophila* clade, which may be one of the best examples of an adaptive radiation in nature and contains about a fifth of the species in the family [12–14], as well as many lesser-studied species or groups that may provide important context to the currently known evolutionary history of drosophilids.

Correcting the taxon sampling bias in genome assemblies is an important step on the path towards a comprehensive genomic study of family Drosophilidae [7,11], but there are technical challenges to address. Establishment of a laboratory culture is a typical first step of generating a drosophilid genome due to the comparatively demanding sample input requirements for long-read sequencing. Recommended genomic DNA (gDNA) inputs range from a few hundred nanograms to several micrograms of high-quality, high molecular weight (HMW) gDNA, often exceeding the amount that can be extracted from an average-sized or smaller drosophilid. Pooling multiple wild individuals increases total gDNA yield but increases the number of haplotypes in the data, leading to inflated consensus error rates and lower assembly contiguities [8,15,16]. This is further exacerbated by the high genetic diversity of many insect populations, as well as issues arising from pools containing mis-identified individuals of similar species.

While single-specimen insect genome assembly methods [17,18] circumvent the laboratory culture step and can address the taxonomic sequencing bias, the utility of Nanopore sequencing approaches for this purpose has not been thoroughly explored. It is indeed possible to obtain high-quality genome assemblies from single small organisms like drosophilids [17,19]; however, the aforementioned low-input assembly protocols typically increase the material for library construction by employing some form of amplification, and offset contiguity decreases from shorter reads by scaffolding contigs with Hi-C. These methods, while effective on a genome-by-genome basis, can introduce amplification bias but more importantly increase costs and add complexity to the process. Applying these methods at the scale of hundreds or thousands of genomes is not currently feasible without the resources of a large consortium. Furthermore, these methods still require fairly fresh (ideally, flash-frozen) samples. Collections held by field biologists, systematists, and museums are a largely untapped resource for addressing taxonomic bias in available genomes, but the efficacy of long-read sequencing for these types of specimens is similarly untested. We therefore sought out both freshly collected and ethanol-preserved (up to 2 decades old) specimens to test the limits of amplification-free Nanopore sequencing approaches for single flies.

Here, we present the outcome of that work and another major step towards unbiased comprehensive genomic study of an entire insect family: 183 new genome assemblies of 179 drosophilid species representing many of the various genera, species groups, and subgroups across Drosophilidae. With this study, genome sequences of 360 drosophilid species (plus 1 new outgroup genome from Family Diastatidae) are available for public download. Of just the new genomes, 121 are from single adult flies and 62 are from laboratory strains. Genomes were predominantly sequenced with a hybrid ONT R9.4.1 (87 genomes) or R10.4.1 (80 genomes) and Illumina approach, except for a small fraction of samples where sample quality issues or low gDNA yield limited our sequencing. The new data include several R10.4.1 runs for community testing and benchmarking, including 378× depth of coverage of the *D. melanogaster* reference [1,20] strain. We present an updated wet lab protocol and genome assembly pipeline that work for sequencing even the most miniscule of the drosophilids (e.g., some *Scaptomyza* spp. individuals that we estimate to be less than 1/2 the length of adult *D. melanogaster* and that yield

35 ng gDNA from 1 adult) at a per-sample cost of US \$150 including short reads. Even at this lower limit, our single-fly sequencing approach regularly produces contiguous (>1 Mb contig N50), complete (>98% BUSCO), and accurate (>QV40 genome-wide with ONT R10.4.1) genomes. Finally, we present a reference-free, whole-genome alignment of 298 drosophilid species to facilitate comparative genomic studies of this model system.

## Results and discussion

### Taxon sampling

We selected additional species for sequencing with the primary objective of improving the taxonomic diversity of genomes of species across the family Drosophilidae (**Fig 1**). Robust inference of historical evolutionary relationships among species and higher taxonomic groups is a key first step that lays the foundation for future study into drosophilid evolution. To date, the largest molecular phylogeny of the group is based on 17 genes from 704 species [7]. While these data are by far the most comprehensive in the number of species surveyed, many deep branches in the phylogeny and many of the exact relationships of species within species groups and subgroups are not confidently resolved. More loci are needed to resolve the species tree, particularly in the presence of extensive introgression and incomplete lineage sorting [21]. Whole-genome sequencing, particularly long-read sequencing, makes thousands of orthologous loci immediately accessible and addresses these issues.

Sampling was conducted across the family as follows. From the TaxoDros database [22], family Drosophilidae is split into the lesser studied subfamily Steganinae, for which we sequenced 9 species from 5 genera (*Stegana*, *Leucophenga*, *Phortica*, *Cacoxenus*, and *Amiota*) and the better known subfamily Drosophilinae. Within the subfamily Drosophilinae, we sequenced 8 species from 4 genera (*Colocasiomyia*, *Chymomyza*, *Scaptodrosophila*, and *Lissocephala*) that are outgroups to the large, well-studied, and paraphyletic genus *Drosophila*. Previous studies (e.g., [7,14,23,24]) have long noted this paraphyly, but a taxonomic revision has not occurred in part due to potential effects on the nomenclature of model organisms and due to uncertainty about the placement of many taxa. We therefore sampled 22 species from 14 genera that render the genus *Drosophila* paraphyletic (*Collessia*, *Dettopsomyia*, *Dichaetophora*, *Hirtodrosophila*, *Hypselothyrea*, *Liodrosophila*, *Lordiphosa*, *Microdrosophila*, *Mulgravea*, *Mycodrosophila*, *Phorticella*, *Sphaerogastrella*, *Zygothrica*, and *Zaprionus*). Finally, we sampled new species from the *testacea*, *quinaria*, *robusta*, *melanica*, *repleta*, and Hawaiian *Drosophila* species groups. For the *Scaptomyza*-Hawaiian *Drosophila* radiation specifically, we present 63 new genomes from most major groups, including multiple subgenera of the genus *Scaptomyza* (*Exalloscaptomyza*, *Engiscaptomyza*, *Elmomyza*) and representatives of the *picture-wing*, *haleakale*, *antopocerus*, *modified tarsus*, *ciliated tarsus*, and *modified mouthpart* species groups and their subgroups. We also included 2 species, *D. maculinotata* and *D. flavopinicola*, considered to be close relatives of the *Scaptomyza*+Hawaiian *Drosophila* lineage [7,25].

### Highly accurate genomes with Nanopore R10.4.1 sequencing

Oxford Nanopore sequencing hardware and chemistry have seen major upgrades in the shift to R10.4.1 and are now able to read DNA fragments at >99% single-read accuracy (the "Q20 chemistry"). To assess consensus sequence accuracy of drosophilid assemblies using these reads, we extracted HMW gDNA from male adult flies of the *D. melanogaster* Berkeley *Drosophila* Genome Project iso-1 strain and generated 54.4 Gbp (378× depth) of ONT R10.4.1 data, with a read N50 of 28,261 bp and containing 11.2 Gbp (79× depth) of reads 50 kb in length or longer. About 40× depth of publicly available short-read data [10] were downloaded from NCBI. Using the full dataset, we assembled (Methods) 142.1 Mbp of genomic sequence

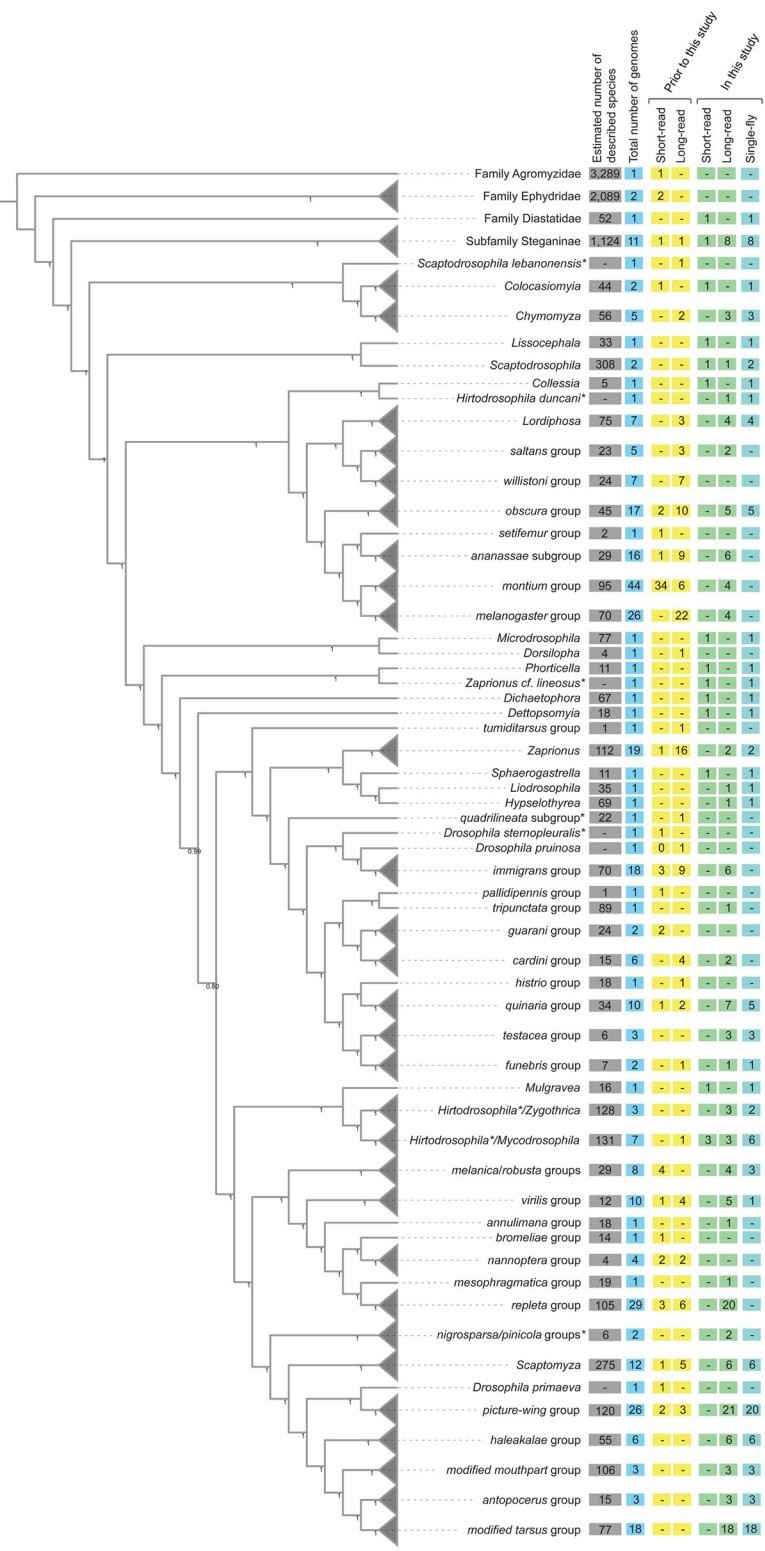

**Fig 1. Cladogram of drosophilid species with whole-genome data, with some groups collapsed (gray triangles).**
Species relationships were inferred from 1,000 orthologs (see Methods). Node values are the local posterior
probabilities reported by ASTRAL-MP [26]. Counts of described species for each group were obtained from the
TaxoDros database [22]. Values in the colored boxes indicate, as of August 2023, the number of species with whole-
genome sequences for each taxon. The count of short-read and long-read datasets, and data available before this study

(including [8]) and new genomes presented here are shown separately. *Note that *Scaptodrosophila*, *Hirtodrosophila*, *Zaprionus*, *immigrans*, and *histrio* groups are potentially rendered polyphyletic by these samples. The positions of *nigrosparsa* and *pinicola* groups are currently considered to be uncertain. The data underlying this figure can be found in https://doi.org/10.5281/zenodo.11200891.

in 93 contigs, with a contig N50 of 21.7 Mbp and a genome-wide consensus accuracy (QV46.0) comparable to the dm6 [27] reference genome (QV43.4). Of all the 17,867 features annotated in the dm6 reference genome, we were able to transfer with LiftOff [28] 17,627 (98.7%) to the ONT assembly, and of the subset of 13,967 protein-coding genes, 13,870 (99.3%) were successfully lifted over to the ONT assembly.

Next, we assessed the performance of R10.4.1 for more practical levels of sequencing coverage. We downsampled the original reads to 10 replicate datasets for each of 20×, 25×, 30×, 40×, 50×, and 60× depths of coverage and ran our genome assembly pipeline both with and without additional Illumina polishing (using all 40× of the short-read data each time). Gene annotations were again lifted from the dm6 reference genome to each new assembly, and consensus quality scores were evaluated for the entire genome, then separately for each of the major chromosome scaffolds, coding sequences, introns, and intergenic regions (**Fig 2 and**

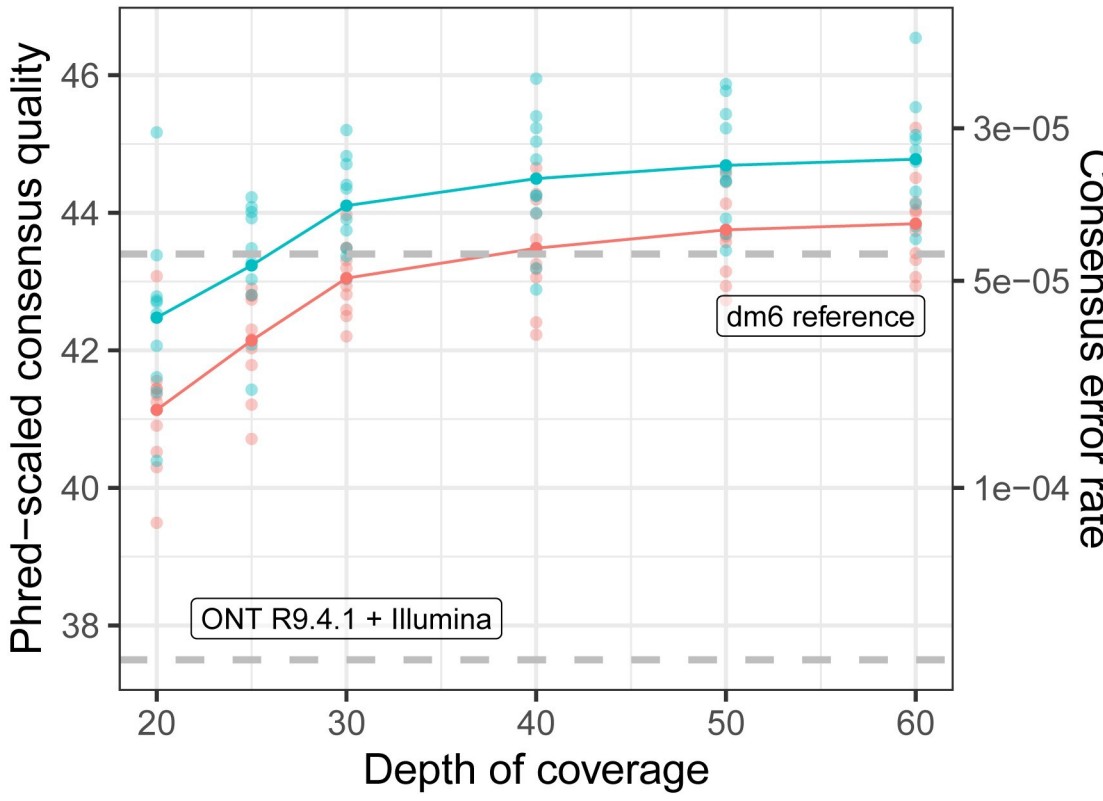

**Fig 2. High genome-wide consensus accuracy with Nanopore R10.4.1 sequencing.** The Phred-scaled consensus accuracy (left axis) and per-base consensus error rate (rate) are shown for genomes built with 20× to 60× coverage of ONT reads. Dashed gray lines show consensus accuracy estimates for R9.4.1 + Illumina [8] and the dm6 reference genome [20]. The data underlying this figure can be found in https://doi.org/10.5281/zenodo.11200891.

**S2** and **S3** Tables). To assess which genomic regions were best and most consistently assembled, we mapped each assembly to a reference genome and computed alignment coverage over each major chromosome (**S2 Fig**). The *D. melanogaster* Y chromosome (estimated to be approximately 40 Mbp) is composed of repeat-rich heterochromatin and is poorly assembled, even in the current dm6 reference assembly (approximately 4 Mbp). We used an alternative, heterochromatin-enriched assembly with an additional 10.6 Mbp of Y-linked sequences [29] for these reference alignment-based assessments, reasoning that it would provide a better, albeit still limited, evaluation of the completeness of the Y chromosome.

Even for lower ($<30\times$) coverage ONT-only datasets, genome-wide consensus accuracy (**Fig 2**) exceeds both the standard recommended by the Vertebrate Genomes Project (QV40) [30] and the genome-wide consensus accuracy we previously reported (QV37.15) for a *D. melanogaster* R9.4.1 and Illumina hybrid assembly [8]. Consensus accuracy is fairly stable past $30\times$ depth but increases in total by about QV1 from $30\times$ to $60\times$ depth of coverage (**Fig 2**). As we previously observed, consensus accuracy varies across both chromosomes and genomic elements (**S2** and **S3** Tables), where autosomes are more accurate than sex chromosomes and coding sequences ($>$QV55 at $\geq30\times$ depth) are significantly more accurate than the rest of the genome. Of the ~23 Mbp of coding regions in the *D. melanogaster* genome we assembled with $60\times$ ONT data, we detect an average of 47 errors (or QV56.9) for R10.4.1-only assemblies and 32 errors (or QV58.6) for Illumina-polished assemblies. For most genomic elements, additional polishing with Illumina data slightly improved consensus accuracy by ~QV1-2 except for contigs mapping to the dm6 Y chromosome, which were not improved by short-read polishing.

The depth of ONT sequencing coverage had little impact on assembly completeness for the major *D. melanogaster* chromosomes (S2 Fig), roughly following the patterns exhibited by the consensus accuracy estimates. The major euchromatic chromosome arms (2L, 2R, 3L, 3R, 4, and X) were well assembled and exhibited similar high degrees of completeness (all approximately 90% or above) across the entire range of downsampled coverages, even though we expected about half coverage on the sex chromosomes relative to the autosomes (e.g., $10\times$ X/Y versus $20\times$ autosome) from male flies. Similarly, the Y chromosome was always poorly assembled (about 10% complete) irrespective of coverage. While this result is expected given previous efforts to assemble the Y chromosome [27,29], our results further indicate that a modest increase in read lengths (approximately 28 kb read N50 in this study versus approximately 14 kb read N50 in [29]) and increasing sequencing coverage have little effect on improving assembly quality, particularly for repeat-rich heterochromatic sequences. More optimistically, these results demonstrate the effectiveness of even modest long-read datasets for assembling the majority of the genome.

While we will not delve further into assembly of the *D. melanogaster* reference genome here, our benchmarking clearly shows the R10.4.1 ONT and Illumina hybrid approach is a cost-effective way to generate high-quality assemblies that meets current standards for reference genomes. We also note a significant increase in R10.4.1 chemistry sensitivity over R9.4.1, in other words, that about 1/5 to 1/10 (by mass) of loaded library is needed to achieve similar pore occupancy to R9.4.1 for libraries of the same fragment size distribution. This indicates great potential for low input ONT library preps and is the critical factor that makes it possible to sequence the more challenging samples presented shortly.

Despite the relatively small improvements to sequence accuracy provided by short-read polishing, we still recommend Illumina sequencing for Nanopore-based genome assembly projects of drosophilid species. The input requirements (1 to 10 ng gDNA) and the per-genome costs of Illumina sequencing (US \$35 for $30\times$ coverage of a 200 Mbp fly genome) are minimal. Short-read data provide an avenue for assessing assembly accuracy in a less biased

reference-free manner [31,32] and are easier to integrate into downstream population genomic analyses within a single variant calling pipeline for genomes from wild-collected individuals.

## Haplotype phasing improves the accuracy of single-fly assemblies

One of the recent major developments in genome assembly is the class of methods that consider linkage information contained in long reads from a diploid sample to construct a phased diploid assembly, rather than a single haploid assembly. Diploid assembly methods can incorporate phasing during assembly [31], phase while polishing a haploid draft genome [15,33], or correct long reads prior to genome assembly in a haplotype-aware manner [34]. These methods all aim to minimize issues arising from random haplotype switches in a haploid consensus of a diploid organism, which may have downstream effects on read mapping, variant calling, and out-of-frame errors in coding regions [15]. Further, switch errors between variants in close proximity may introduce novel k-mers into the reference genome that will be counted as errors. While these genomes are not completely phased with respect to the parents and may still randomly switch parental phase within contigs, we find that the diploid assembly approach (Methods) improves assembly consensus accuracy by up to an order of magnitude (from QV29.7-QV37.3 to QV40.3-QV58.2) in a head-to-head comparison with a subset of our single-fly genome samples (**Fig 3**). Variants called separately with Illumina and ONT reads are in general highly (>90%) concordant for 90 out of 101 tested samples (S4 Table), further indicating the effectiveness of even modestly long reads for variant calling and phasing here. These results also imply that diploid assembly of single flies is a superior strategy to assembly from pools of wild individuals. Phased diploid assembly of single flies will therefore be the primary strategy for genome assembly with wild-collected flies in this manuscript and in future work.

## 183 New drosophilid whole-genome sequences

Here, we present 183 new genomes for 179 drosophilid species. Of this total, 163 genomes were assembled with a hybrid ONT and Illumina approach, 4 were assembled with only ONT R10.4.1, and 16 were assembled only from Illumina paired-end reads. Sixty-two genomes were assembled with material from a laboratory stock, and the other 121 genomes were assembled from material extracted from a single adult fly. As described, these data improve the depth of sampling for key taxa, such as the Hawaiian *Drosophila*, multiple mycophagous taxa, and the *repleta* group, but also capture at least 1 species from many, but not all, drosophilid clades without a sequenced representative (**Fig 1**).

High-quality genome assemblies were generated from these data following our pipeline for haploid assembly from laboratory lines [8], or a diploid assembly workflow for single, wild-caught flies (see Methods for details). Most of the long-read assemblies are highly contiguous, complete, and accurate (**Fig 4**), in most cases exceeding the QV40 and 1 Mb contig N50 minimum standards proposed by the Vertebrate Genomes Project [30]. We note that genome-wide sequence quality for R9.4.1 drosophilid hybrid assemblies usually does not exceed QV40 even though we have previously demonstrated coding sequences in R9.4.1 hybrid assemblies to be highly accurate [8]. The full details on the samples underlying these assemblies and the corresponding genome QC metrics are provided in **S4 Table**.

The major factors limiting assembly quality were contamination and sample quality. Freshly collected samples had minimal issues during the library prep steps, but microbial (e.g., bacterial, nematode) sequences were abundant in wild-collected specimens (particularly in *quinaria* group), reducing on-target read coverage of both Illumina and Nanopore reads and thus genome contiguity, accuracy, and concordance of SNPs called separately with both types of reads. For some of these samples (specifically, *D. subquinaria*, *D. suboccidentalis*, *D. recens*,

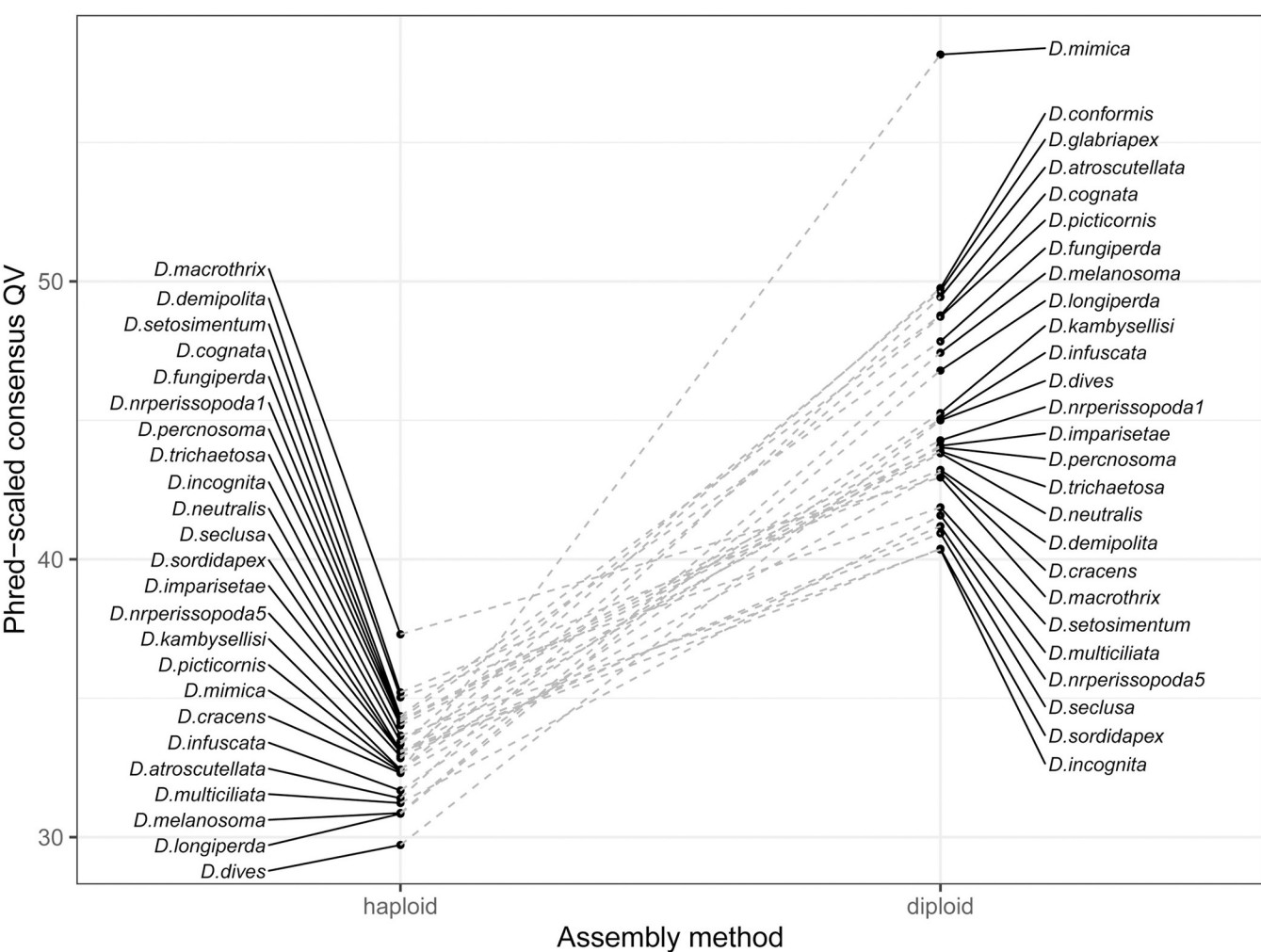

**Fig 3. Consensus accuracy for single-fly genomes is greatly improved by diploid assembly.** Phred-scaled consensus quality (QV) is shown for a subset of 25 R10.4.1 and Illumina hybrid single-fly genomes assembled with haploid (left) and diploid (right) pipelines. The data underlying this figure can be found in https://doi.org/10.5281/zenodo.11200891.

and *D. rellima*), we made multiple attempts at a single-fly genome assembly and present the best one here. Older ethanol-preserved specimens were also challenging to prepare and assemble. Heavily fragmented and degraded gDNA (e.g., *Zaprionus obscuricornis* and *Phortica magna*) limited our ability to remove shorter gDNA fragments with size selection buffers during the ONT library prep. The co-purification of unknown contaminants (frequently present in some of the older Hawaiian picture-wing flies like *D. quasianomalipes*) required us to perform additional sample purification steps that led to significant sample loss. All these factors negatively impacted read throughput and consensus quality. Genome quality metrics reported in **S4 Table** are generally lower for these samples.

Limited yield and/or highly fragmented gDNA limited us to Illumina sequencing for 16 specimens. The lack of Nanopore data for these gDNA-limited specimens is noted in **S4 Table**. They tended to be older (collected between 2008 and 2012, with the exception of *P. flavipennis* in 2017) and smaller flies and were all stored long term in absolute ethanol. For these datasets, a simple draft assembly was generated from paired-end reads with only a contaminant removal post-processing step. While the utility of the latter type of assembly for

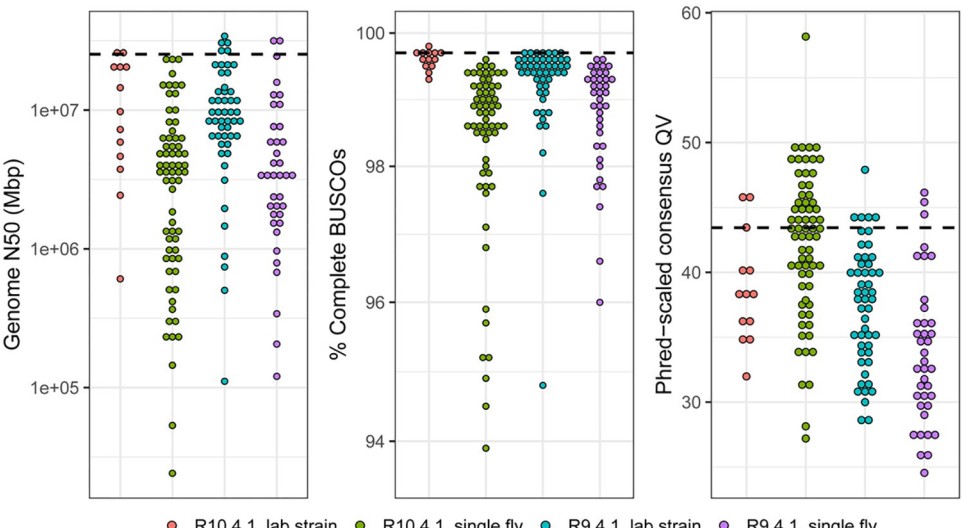

**Fig 4. The distribution of genome quality metrics for 168 new long-read assemblies.** Distributions of genome N50, the percentage of complete dipteran BUSCOs [35], and Phred-scaled QV are plotted separately for R10.4.1 and R9.4.1 assemblies from lab strains and from single flies. The black dashed line is the value computed for the *D. melanogaster* dm6 reference genome. The 16 samples that were only sequenced with Illumina are omitted from these plots. Data underlying this figure and additional sample information is provided in **S4 Table**.

population or comparative genomics is limited, the sequences are useful for phylogenetic inference [36,37].

## Comparative resources based on whole-genome data

We present a number of additional resources to assist users of these data with downstream comparative genomic analyses. First, we provide a curated (current as of August 2023) list of representative whole-genome datasets of 360 drosophilid species and 4 outgroups (1 agromy-zid, 2 ephydrids, and 1 diastatid) in **S1 Table**. This table provides a convenient reference for readers to keep track of the genomes incorporated into the resources we present. This can be especially challenging in such a rapidly changing field. Updated versions of this list will be presented in future work.

We inferred species relationships of these 364 genomes using 1,000 dipteran BUSCO genes [35,38] identified as complete and single copy across the most genomes (**S1 Fig**). As expected, the relationships of the major species groups in our phylogeny remains mostly consistent with previous work [7,24]. The differences also reflect our much larger set of orthologs: deep-branching relationships between the clade containing *Mulgravea*, *Hirtodrosophila*, *Zygothrica*, and *Mycodrosophila*; *Dichaetophora*; *Dettopsomyia*; and the *Drosophila* and *Siphlodora* subge-nera are confidently resolved, as well as the more recent evolutionary relationships between nearly all individual species. Interestingly, we still observe some uncertainty (local posterior probability <0.9) for a few branches in the phylogeny. The cause of increased gene tree-species tree discordance at these branches is currently unknown but will be investigated using a complete set of orthologous markers in future work. A more detailed discussion of the taxonomic implications of whole-genome sequencing will be reserved for this forthcoming study. Lastly, the phylogeny was scaled by the substitution rate at 4-fold degenerate sites in the BUSCO genes to provide a guide tree for whole-genome alignment.

A Progressive Cactus [39] reference-free whole-genome alignment was computed for a subset of 298 genomes (**S5 Table**) out of the 364 species listed in **S1 Table**. The Cactus alignments and

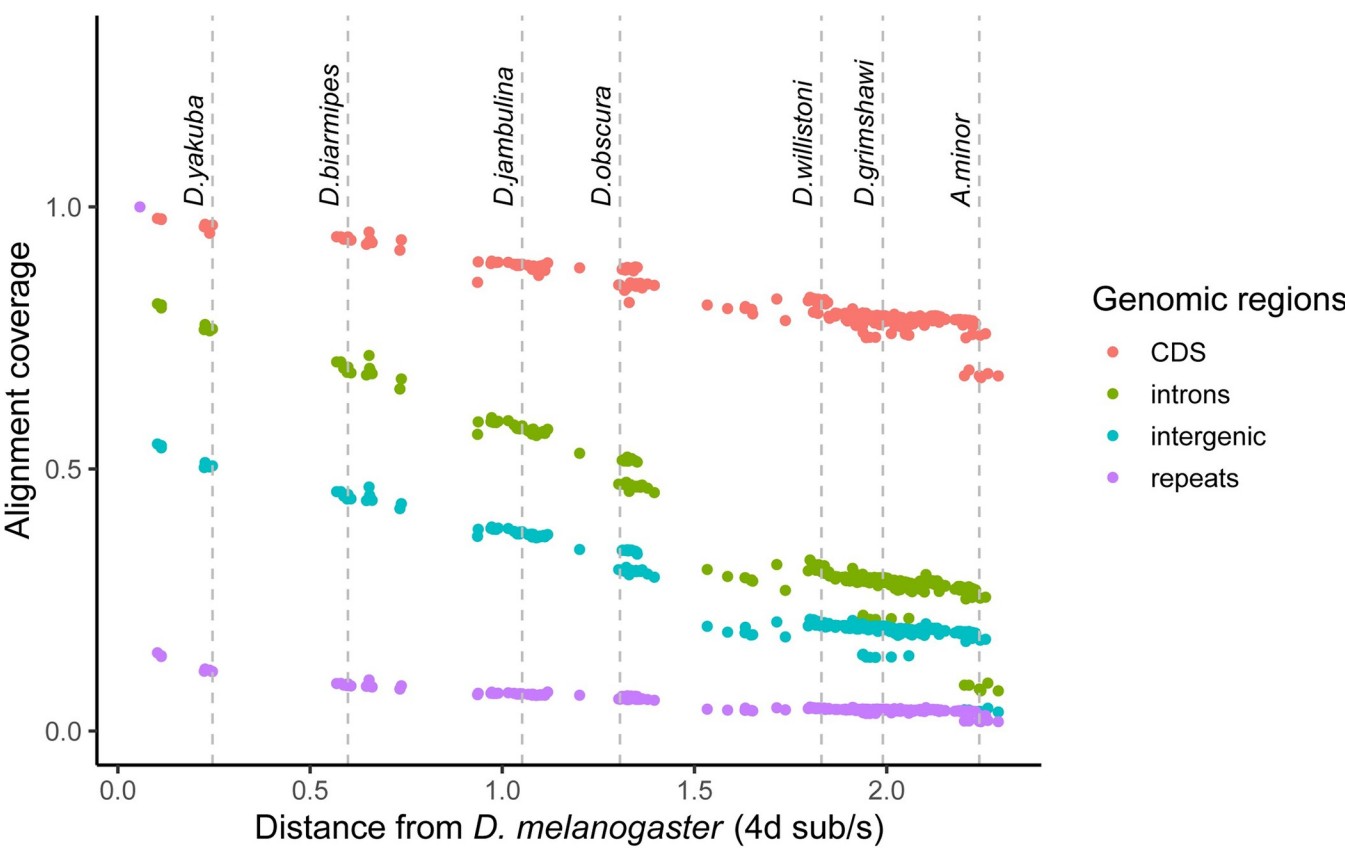

**Fig 5. Proportion of *D. melanogaster* genomic elements aligning to other species as a function of 4-fold divergence from *D. melanogaster*.** Each dot represents 1 species. Alignment coverage is defined as the proportion of genomic elements in *D. melanogaster* that uniquely map to another species. The data underlying this figure can be found in https://doi.org/10.5281/zenodo.11200891.

HAL file format utilities [40] are part of a large suite of comparative genomics tools that allow users to quickly perform liftover of genomic features, comparative annotation [41], compute evolutionary rate scores [42], and more. To minimize issues from low-quality assemblies, genomes for the alignment were selected based on minimum contiguity (N50 > 20 kb) and minimum completeness (BUSCO >95%) filters. For species with multiple assemblies present, we deferred to the NCBI representative genome unless our new genome was a major improvement. Alignment coverage of *D. melanogaster* genomic elements in other species is depicted in **Fig 5**, showing that most of the protein-coding genome aligns across Drosophilidae. The inferred ancestral drosophilid genome is 33.5 Mbp in size, about 10 Mbp larger than the sum of *D. melanogaster* coding sequences and contains 97.3% of the dipteran BUSCO genes as complete and single-copy. This suggests the ancestral assembly is enriched for functional sequences and provides an upper bound for the total amount of functional sequence conserved across drosophilid genomes. Moreover, the estimated average substitution rate at 4-fold degenerate sites is 44.8 substitutions/site (**S1 Fig**), suggesting a complete saturation of substitutions at every neutrally evolving site in the alignable genome and base-level resolution of comparative genomics approaches for evolutionary rate estimation [43]. Together, these results demonstrate the immense potential for comparative genomics provided by this dataset even across deep evolutionary timescales. We note that fitting branch lengths with 4-fold sites alone may mis-estimate deeper branch lengths due to substitution saturation and will reexamine this problem with more complex codon substitution models in forthcoming work on the Drosophilidae Tree of Life.

## Remaining challenges

As another major step towards clade-scale genomics of the family Drosophilidae, we have demonstrated that assembling genomes from individual flies, even those preserved in ethanol for up to 2 decades, is now feasible. We are still far from sequencing every, or even most, species in the family. We are faced with several immediate challenges as this resource grows, and it is important for users to be aware of them.

There is significant variation in genome quality and completeness despite the generally optimistic genome quality metrics we have reported. Read lengths for single-fly library preps are usually around an order of magnitude shorter than those from inbred lines due to sample material limitations. This difference is particularly egregious for the most fragmented gDNA extractions from the oldest ethanol samples, which are close to the lower limits of what is acceptable for ONT sequencing and assembly (1,000 bp read N50). The reconstruction of repetitive sequences and other complex genomic regions will be severely limited from such short sequences, although these data are still superior to paired-end short reads for genome assembly. We urge that comparative genomic analyses, particularly those of structural variation, carefully consider these possibilities and do not readily consider absence of sequence as evidence of absence.

As this resource grows beyond the most commonly studied drosophilids, ambiguity in species identification and taxonomy may become significant issues. All wild-collected samples are, to the best of our ability, identified through key morphological characters prior to sequencing [44,45]. It is nevertheless inevitable that we will misidentify or sequence ambiguous specimens, especially ones that belong to rare and/or cryptic species (e.g., *D. colorata*) or those with historically inconsistent taxonomic placement (e.g., *D. flavopinicola*). In a few cases, we have discovered through sequencing that even laboratory and stock center lines were misidentified or contaminated (usually by *D. melanogaster*, see Materials and methods). We try to minimize these issues by checking genome assemblies against known marker sequences on the Barcode of Life [46] and NCBI Nucleotide databases (Methods), but even then there are many species, some not formally described, for which basic genetic markers are not readily available. It is possible for these markers to come from incorrectly identified samples too. Whole-genome sequencing approaches applied alongside comprehensive field collections will help address the majority of these taxonomic and data quality problems. We will continue to update and correct records as new datasets are generated or if errors in identification are detected.

## Next steps

In creating these resources, we aim to build a powerful open resource for drosophilid evolutionary genomics and ultimately a framework for connecting micro- to macro-evolutionary processes in Drosophilidae and beyond. This study is just one part of an ongoing set of community-level projects developing this genomic resource. We will briefly describe ongoing efforts to inform readers about additional new resources that will be available in the near future.

In addition to continuously assembling new genomes, we are working on improvements to existing genomes. We are generating new R10.4.1 data with the specific intent of improving consensus accuracy for existing R9.4.1 assemblies along with new transcriptomic data for gene annotation. For species with lines readily available from the National *Drosophila* Species Stock Center, heads and bodies from pools of adult males and females (4 libraries per species) will be sequenced. For wild-collected specimens, libraries will be prepared from single whole adults. Full-length transcripts from select species will be sequenced with long-read approaches. Importantly, we will submit sequences for annotation by NCBI RefSeq to maintain consistency in gene annotation pipelines [47].

A well-resolved phylogeny is crucial for comparative analyses, but there are many open phylogenetic and taxonomic issues to be resolved in Drosophilidae. While it is clear that whole-genome sequencing is the best way to infer a reliable phylogeny, there are legacy Sanger datasets for hundreds more species from many past molecular phylogenetic studies of this group [7,11] that remain useful for inference of the Drosophilidae Tree of Life. The whole-genome data we are generating will provide both a comprehensive topology for a broader phylogenetic analysis that includes classic marker data, and a way to connect disparate markers from past studies using complete genomes.

Until now, our sampling efforts have been focused on readily available specimens and easily accessible populations of drosophilids. There are still large geographical gaps in the data, particularly in Africa, South America, and Australia—all regions that host significant fractions of global drosophilid biodiversity. Efforts to collect in these regions will be especially fruitful if in partnership with local field biologists—collaborations that we are making efforts to establish now. Even among new groups sampled here, such as the Hawaiian drosophilids, there are major lineages (e.g, the *Scaptomyza* subgenus *Titanochaeta*, and representatives of the *nudidrosophila*, *modified mouthpart*, and *rustica* species groups) that remain to be sampled. Museum specimens, like the ethanol ones presented here, but also dry-pinned specimens [48], are proving to be viable material for drosophilid genomics, potentially even providing a valuable look into the past genomic diversity of many species.

Whereas drosophilid species, especially *D. melanogaster*, have played an important role in the development of population genetics, they have also served as early subjects of comparative population genetic studies [49–53], that is, the study of population genetic processes across species. The family Drosophilidae is uniquely well positioned as a system to accelerate the development of this growing field, and the tools provided in this manuscript provide a coherent framework for comparative population genomics at the scale of entire large clades. We are working on polymorphism data from wild populations for many (100 s) of the genomes featured here that will be presented in a forthcoming study of population genomic variation across the entire family. The relationship of population genomic variation with broad-scale macro-evolutionary patterns such as species diversity is still largely unknown [54] and this will be a major step towards addressing this fundamental question in evolutionary biology.

## Materials and methods

### Fly collection

Adult flies were collected for sequencing from laboratory cultures (63 species) or from field collections (101 species). Laboratory strains were grown on species-appropriate media. Mature adults were collected, sexed, and starved for 1 day before dry collection at −80°C or into ethanol. Wild individuals were obtained from older ethanol collections or fresh through collection in the field. Field samples were obtained from banana, mushroom, and watermelon baits and by sweep netting, then stored in ethanol, RNA later, or RNA/DNA Shield (single samples) depending on sample purpose and transport method for the samples. Once in the lab, all samples were promptly stored at −20/−80°C until sequencing. Collection details are listed in **S4 Table**. Older ethanol-fixed specimens were obtained from collections held by M.J. Toda and D.K. Price.

### Ethics statement: Field research

Specimens were collected and/or maintained under the following permits. To S.H. Church: (1) National Parks HAVO-SHC-2022-SCI-0022; (2) DOFAW Native Invertebrate Research Permit with NARS I5081; (3) Kaua'i Forest Reserves KPI-2022-289; (4) Kaua'i State Parks

DSP-KPI-2022-288; (5) Hawai'i Forest Reserves Access Permit (no number issued). To J. Hrcek: (1) WITK16977516 and (2) PTU20-002501 from Queensland Government. To A. Kopp: (1) USDA APHIS P526P-22-06553 and (2) P526P-20-02787. To D. Matute: Permit granted by the São Toméan and Zambian authorities. To M.J. Medeiros: (1) Hawai'i State Department of Land and Natural Resources I5297 and (2) Kaua'i Access Permit KPI-2022-298. To D. Obbard: land-owner permissions granted by email from Keith Obbard, Hugh Gibson, Effie Gibson, Sandy Bayne, and City of Edinburgh Council. To M.J. Toda: (1) Economic Planning Unit of Malaysian Government (research permissions UPE: 40/200/19 SJ.732 and UPE: 40/200/19 SJ.1194 and 1195) and (2) Ministry of Research and Technology of Indonesia (research permissions: 5816/SU/KS/2004, 6967/SU/KS/2004, and 416/SIP/FRP/SM/XI/2013). To T. Werner: (1) USDA Forest Service, Custer Gallatin National Forest (file code 2720); (2) State of Idaho IDFG Wildlife Bureau #764-22-000052; (3) United States Department of the Interior, National Park Service, Olympic #OLYM-2022-SCI-0033; (4) United States Department of the Interior, National Park Service, Olympic #OLYM-2023-SCI-0023; (5) USDA Forest Service, Umpqua National Forest (file code 2720). To T. Werner and B.Y. Kim: State of California, Natural Resources Agency, #23-635-014.

## Genomic DNA extraction and library prep

The gDNA extraction and library prep methods employed here follow the protocols described in our previous work [8], with slight simplifications to streamline the process. Ethanol-fixed flies were rehydrated by 30 min of incubation in Buffer STE (400 mM NaCl 20 mM; Tris-HCl (pH 8.0); 30 mM EDTA). Rehydrated or frozen flies were crushed with a plastic pestle and immediately mixed with warmed 200 to 500 µl lysis buffer (0.1 M Tris-HCl (pH 8.0); 0.1 M NaCl; 20 mM EDTA; 250 µg/ml proteinase K; 0.6% SDS) and incubated at 50 to 60°C for up to 4 h. RNAse A was added to a final concentration of approximately 190 µg/ml (about 1 µl of 20 µg/ml stock per 100 µl lysis buffer) 30 min before phenol-chloroform extraction. The lysate was transferred to a phase lock gel tube and subjected to 2 phenol-chloroform and 1 chloroform extraction. The aqueous layer was decanted to a new 1.5-ml tube and DNA precipitated by the addition of 10% 3 M NaOAc and 2 to 2.5 volumes of cold absolute ethanol. The DNA was pelleted by centrifugation, resuspended in 26 µl 10 mM Tris (pH 8.0) at 4°C overnight or until homogeneously resuspended, then quantified by Nanodrop and Qubit. One to 3 aliquots of purified gDNA (a minimum of 20 ng unless sample limited) were reserved for Illumina library prep and the rest (range 30 to 2,000 ng of gDNA) taken for Nanopore sequencing.

Nanopore libraries were prepared with the ONT ligation kit workflow (SQK-LSK110 for R9.4.1 flow cells and SQK-LSK114 for R10.4.1 flow cells), reducing reactions to half volumes from the ONT recommendations at all steps to reduce library prep cost. We do not observe reduced library yield or lower sequencing throughput from this reduction to the library prep volumes. If starting with gDNA concentrations of >50 ng/µl, we performed DNA fragment size selection with the PacBio SRE kit prior to starting the prep. The DNA repair and end-prep incubation steps were increased to 60 min at 25°C and 30 min at 65°C from the standard 5 min for each step. Elution steps during bead cleanups were extended to 30 min or longer (4 to 6 h) on the heat block at 37°C or until beads were homogeneously resuspended. The adapter ligation step was also increased to 30 to 60 min from the standard 5 min. Prepared library was sequenced on an R9.4.1 or R10.4.1 flow cell according to the manufacturer's instructions with live basecalling on in fast mode. Flow cells were washed in between sequencing runs with the ONT EXP-WSH004 flow cell wash kit, following the manufacturer's instructions.

For Illumina library preps, 5 to 10 ng DNA in 6 µl were prepared with the Illumina DNA library preparation kit. We followed the standard workflow but reduced reaction volumes by

1/5 as a cost-saving measure. Libraries were amplified with 6 PCR cycles, except for the most sample-limited specimens, which were amplified with 7 PCR cycles (with more sequencing coverage to compensate for the increased duplication). Prepared libraries were individually cleaned up with beads and groups of samples with comparable genome sizes pooled by ng/μl concentration. Fragment size distributions were checked with BioAnalyzer, and the molar concentration of library pools were quantified by qPCR in-house, by the Stanford PAN facility, or by Admera Health. Sequencing was performed on NovaSeq S4 and NovaSeq X Plus machines at the Chan-Zuckerberg Biohub (San Francisco, California, United States of America) and Admera Health (South Plainfield, New Jersey, USA).

A detailed step-by-step sequencing protocol is available at Protocols.io and as a spreadsheet in the Supplementary Materials.

## Sequencing cost estimate

The cost of sequencing is one of the major limiting factors for large genome assembly projects like this study. Here, we have highlighted an estimated sequencing cost of USD $150 per sample as a benchmark that reflects both improvements in protocols and the sequencing technology. The specifics of this estimate are provided in S6 Table. Note that the per-sample costs may differ from this estimate based on available sequencing resources and the scale of the project.

## Genome assembly pipeline

ONT raw sequencing data were processed with standard tools. After initial sequencing, reads were basecalled with Guppy 6 and the appropriate (for R9.4.1 and R10.4.1) super-accuracy basecalling model with the adapter trimming and read splitting options on. Adapter trimming was performed on Illumina data with BBtools [55].

Two separate genome assembly pipelines were executed for inbred lines and wild-collected flies. Inbred lines were assembled with a haploid assembly pipeline. ONT reads were assembled with Flye [56] and haplotigs were identified and removed from the draft assembly with 1 round of purge_dups [16]. The assembly was then polished using Nanopore data with 1 round of Medaka and further polished using Illumina data with 1 round of Pilon [57] with only base-level corrections enabled for the latter. Finally, contaminant sequences were flagged and removed with NCBI Foreign Contamination Screen (FCS, [58]). Single wild-caught flies were assembled with a diploid assembly pipeline. Haplotype-aware correction was applied to ONT reads using Illumina data with Ratatosk [34], corrected reads were assembled with Flye, contaminant sequences flagged and removed with NCBI FCS, haplotigs identified and removed by 1 round of purge_dups, and a phased dual assembly was generated with Hapdup [33]. Repetitive sequences were identified in both haploid and the primary diploid assemblies with RepeatModeler2 [59]. Sequences were soft-masked with RepeatMasker [60], using the genome-specific repeat library. The *D. miranda* genome was scaffolded into chromosomes with RagTag [61], using the current NCBI RefSeq genome [62] as a reference. In this specific case, we ignored the NCBI RefSeq genome due to extensive BUSCO duplication in the assembly.

Genome quality metrics were assessed with a number of tools. Genome contiguity statistics were computed with GenomeTools [63]. Completeness of protein-coding genes was assessed with BUSCO v5 [35] and the OrthoDB diptera_odb10 set [38]. If Illumina reads were available, genome accuracy (QV) was assessed with the reference-free, k-mer–based approach implemented in Yak [31]. If only Nanopore reads were available, consensus QV was assessed with a mapping and variant calling approach [8,10]. Nanopore reads were mapped to the draft assembly with minimap2 [64] and variant calling was performed with PEPPER-Margin-DeepVariant

[65]. Homozygous non-reference genotypes were considered to be errors and QV was computed as QV = $-10*\log_{10}$(number of homozygous derived/number of callable sites). Reference-based completeness evaluation was performed by mapping a test assembly against a reference with minimap2 (with secondary alignments off) and computing the breadth of coverage for each major chromosome in the reference.

### Short-read assembly

Genomes were assembled from short-read only datasets with minimal additional processing. Paired-end reads were assembled with SPAdes [66]. Similar to long-read assemblies, contaminant sequences were identified and removed with NCBI FCS.

### Additional quality control

We performed further quality control steps in addition to the standard assembly workflow to minimize issues related to incorrectly sexed flies and species misidentification.

To verify fly sex, we leveraged the observation that drosophilid genes rarely translocate across Muller elements [8,67] and reasoned that single copy BUSCO genes on the X chromosome (typically Muller A in drosophilids and in *D. melanogaster*) should have half read coverage of autosomes in sequencing data of males and equal coverage in female reads. We note that sex chromosome transitions are fairly common in dipterans [68], but these transitions typically involve chromosome fusions in drosophilids and are unlikely to completely mislead coverage-based sex verification as a heuristic approach. We created a list of the X-linked BUSCO orthologs on the *D. melanogaster* X chromosome, then for each test genome, examined the depth of ONT read coverage over the same orthologs identified as single-copy and complete in our other assemblies.

To verify species identities, we checked each genome assembly against known marker sequences on the NCBI Nucleotide database and COI sequences on the Barcode of Life Database [46]. Because purge_dups flags contigs in the assembly with abnormal levels of coverage, we performed species verification on the unprocessed Flye draft assemblies. COI and other marker sequences were identified in each genome by a BLAST search with the *D. melanogaster* COI as the query sequence, then extracted sequences were provided to the NCBI BLAST and BOLD web interfaces to search for close matches. We considered markers with 98% sequence identity as positive identifications but recommend caution here as it is possible that public databases may also contain sequences from misidentified species. Finally, individual species' phylogenetic positions were checked against a large published phylogeny [7]. We will continue to correct any additional issues we identify in the future.

For assessment of variant call concordance between Illumina and ONT variant calls in the diploid assemblies, we restricted the comparisons to SNPs and to regions callable with Illumina data. Sites overlapping with repetitive elements or with site-level quality scores (obtained from the gVCF) less than 20 were masked. The intersection of biallelic SNPs called separately with the Genome Analysis Toolkit 4 [69] and PEPPER-Margin-Deepvariant [65] was then computed, and per-base pair heterozygosity was estimated by dividing the number of Illumina-based SNPs by the length of unmasked sequences in each genome. Additional details are provided in **S4 Table**.

### Downloading additional genomes

A representative genome for every sequenced drosophilid species was obtained from NCBI, giving priority to genomes designated as representative genomes by NCBI. Sample information and NCBI RefSeq/GenBank/SRA accessions are provided in **S1 Table**.

## Species tree inference from BUSCO orthologs

A species tree was inferred from up to 1,000 single-copy orthologs identified in each genome following our previous workflow [8,21]. Only the primary assembly was used if haplotype-aware methods were utilized for genome assembly. For all genome assemblies including the ones from only short-read data, we ran BUSCO v5 [35] using the diptera_odb10 dataset [38] with Augustus gene prediction [70] on. Coding and amino acid sequences for the 1,000 most complete single-copy genes across all genomes were collated into single FASTA files. Coding and amino acid sequences were aligned with MAFFT [71] and codon alignments were generated by back-translation with PAL2NAL [72]. Individual gene trees were generated from DNA alignments with IQTREE2 [73] with automatic model selection. A species tree was inferred from the gene trees with ASTRAL-MP [26]. Four-fold degenerate codons in the codon alignments were identified using the *D. melanogaster* BUSCO annotations as the reference and 4-d codons and 4-d sites were extracted in sufficient statistics format using the msa_view tool from the PHAST software package [74]. The 1,000 genes were split into 30 subsets and branch lengths of the ASTRAL species tree were re-estimated with the phyloFit tool for each subset, then average branch lengths across all subsets were computed with the phyloBoot tool. Both tools are in the PHAST package. We note that lengths of deep branches in a phylogeny solely based on 4-fold degenerate sites may be mis-estimated due to substitution saturation, and we will address these issues with more complex substitution models in forthcoming work. Trees were plotted with the Interactive Tree of Life web tool at itol.embl.de [75].

## Progressive Cactus alignment

A Progressive Cactus [39] whole-genome, reference-free alignment was built with a subset of 298 out of the 360 whole genomes listed in **S1 Table**. The list of genomes used is provided in **S5 Table**. Genomes were filtered based on contig N50 (minimum 20 kbp) and BUSCO completeness (minimum 95% complete single copy). One representative genome was used for each species. The species tree scaled by the 4-fold degenerate substitution rate was provided as a guide tree.

Alignment coverage was computed by extracting coding sequences, introns, and intergenic regions from the *D. melanogaster* dm6 genome annotations. Repetitive elements were identified by re-masking the dm6 genome with RepeatMasker set to produce GFF repeat annotation output. Regions were converted into BED file format, merged with BEDtools [76], and lifted to target genomes with the halLiftover tool.

## Strain contamination

We have identified some contaminated fly stocks through the course of this work. All but one comes from internal contamination events and so the correct stock was ordered and sequenced. *Scaptodrosophila lativittata* (NDSSC# 11020–0081.00) was obtained directly from the stock center and turned out to be *D. melanogaster*. Data from contaminated stocks are never used.

## Reproducible workflows with Protocols.IO, Snakemake, and compute containers

Reproducibility and transparency are top priorities for this open-science community dataset. Full, detailed protocols are provided through Protocols.io. Genome assembly from lab stocks (DOI: dx.doi.org/10.17504/protocols.io.81wgbxzb1lpk/v1); genome assembly from single flies (DOI: dx.doi.org/10.17504/protocols.io.ewov1q967gr2/v1). A list of laboratory reagents is

provided in **S7 Table**. Genome assembly pipelines are available from GitHub (https://github. com/flyseq/2023_drosophila_assembly). This repository contains instructions for building compute containers that include all programs used for this manuscript as well as Snakemake [77]-based pipelines for genome assembly. Links for access to resources are summarized in **S8 Table**.

## Supporting information

**S1 Table. Representative genome assemblies for 360 drosophilid species and 4 outgroups.**
(XLSX)

**S2 Table. Consensus accuracy by chromosome for *D. melanogaster* BDGP iso-1 reassembly.**
(XLSX)

**S3 Table. Consensus accuracy by genomic element type for *D. melanogaster* BDGP iso-1 reassembly.**
(XLSX)

**S4 Table. Sample and genome quality information on 183 new genomes by this study.**
(XLSX)

**S5 Table. Information on genomes used for the Progressive Cactus alignment.**
(XLSX)

**S6 Table. Breakdown of the per-genome cost of sequencing.**
(XLSX)

**S7 Table. List of laboratory reagents and materials used for this study.**
(XLSX)

**S8 Table. Links and accessions to data produced by this study.**
(XLSX)

**S1 Fig. Multi-locus species tree of 360 drosophilid and 4 outgroup taxa estimated from 1,000 single-copy orthologs (Methods).** Branch lengths are scaled by the substitution rate at 4-fold degenerate sites. Node values are posterior probabilities computed by ASTRAL. The data underlying this figure can be found in https://doi.org/10.5281/zenodo.11200891.
(PDF)

**S2 Fig. Assembly completeness across multiple sequencing coverage depths.** Each gray dot represents the proportion of each major chromosome of the dm6 *D. melanogaster* reference assembly that aligns to a de novo *D. melanogaster* assembly. Each dataset was generated by downsampling the original 378× ONT dataset to lower coverage. Ten replicate genomes were assembled for each coverage depth. Red dots indicate the mean across 10 replicates. The data underlying this figure can be found in https://doi.org/10.5281/zenodo.11200891.
(PDF)

## Acknowledgments

We thank Anna Mácová for assistance with shipping fly samples. Some of the computing for this project was performed on the Sherlock cluster. We would like to thank Stanford University and the Stanford Research Computing Center for providing computational resources and support that contributed to these research results.

## Author Contributions

**Conceptualization:** Bernard Y. Kim, Samuel H. Church, Dmitri A. Petrov.

**Data curation:** Bernard Y. Kim, Hannah R. Gellert, Samuel H. Church, Maaria Kankare, Zeeshan A. Syed, Artyom Kopp, Darren J. Obbard, Patrick M. O'Grady, Donald K. Price, Masanori J. Toda, Thomas Werner.

**Formal analysis:** Bernard Y. Kim, Hannah R. Gellert, Samuel H. Church, Anton Suvorov.

**Funding acquisition:** Bernard Y. Kim, Michael B. Eisen, Daniel Matute, Dmitri A. Petrov.

**Investigation:** Bernard Y. Kim, Hannah R. Gellert, Samuel H. Church, Michael B. Eisen, Artyom Kopp, Daniel Matute, Darren J. Obbard, Patrick M. O'Grady, Donald K. Price, Masanori J. Toda, Thomas Werner, Dmitri A. Petrov.

**Methodology:** Bernard Y. Kim, Hannah R. Gellert.

**Project administration:** Bernard Y. Kim, Dmitri A. Petrov.

**Resources:** Bernard Y. Kim, Hannah R. Gellert, Samuel H. Church, Anton Suvorov, Sean S. Anderson, Olga Barmina, Sofia G. Beskid, Aaron A. Comeault, K. Nicole Crown, Sarah E. Diamond, Steve Dorus, Takako Fujichika, James A. Hemker, Jan Hrcek, Maaria Kankare, Toru Katoh, Karl N. Magnacca, Ryan A. Martin, Teruyuki Matsunaga, Matthew J. Medeiros, Danny E. Miller, Scott Pitnick, Michele Schiffer, Sara Simoni, Tessa E. Steenwinkel, Zeeshan A. Syed, Aya Takahashi, Kevin H-C. Wei, Tsuya Yokoyama, Michael B. Eisen, Artyom Kopp, Daniel Matute, Darren J. Obbard, Patrick M. O'Grady, Donald K. Price, Masanori J. Toda, Thomas Werner, Dmitri A. Petrov.

**Software:** Bernard Y. Kim.

**Supervision:** Bernard Y. Kim, Dmitri A. Petrov.

**Validation:** Bernard Y. Kim, Hannah R. Gellert, Samuel H. Church.

**Visualization:** Bernard Y. Kim.

**Writing – original draft:** Bernard Y. Kim, Hannah R. Gellert, Samuel H. Church, Anton Suvorov, Dmitri A. Petrov.

**Writing – review & editing:** Bernard Y. Kim, Hannah R. Gellert, Samuel H. Church, Anton Suvorov, Sean S. Anderson, Olga Barmina, Sofia G. Beskid, Aaron A. Comeault, K. Nicole Crown, Sarah E. Diamond, Steve Dorus, Takako Fujichika, James A. Hemker, Jan Hrcek, Maaria Kankare, Toru Katoh, Karl N. Magnacca, Ryan A. Martin, Teruyuki Matsunaga, Matthew J. Medeiros, Danny E. Miller, Scott Pitnick, Michele Schiffer, Sara Simoni, Tessa E. Steenwinkel, Zeeshan A. Syed, Aya Takahashi, Kevin H-C. Wei, Tsuya Yokoyama, Michael B. Eisen, Artyom Kopp, Daniel Matute, Darren J. Obbard, Patrick M. O'Grady, Donald K. Price, Masanori J. Toda, Thomas Werner, Dmitri A. Petrov.

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
