## [Editor Report · Decision Letter 0]

11 Nov 2023

Dear Dr Kim, 

Thank you for submitting your manuscript entitled "Single-fly assemblies fill major phylogenomic gaps across the Drosophilidae Tree of Life" for consideration as a Methods and Resources by PLOS Biology.

Your manuscript has now been evaluated by the PLOS Biology editorial staff, as well as by an academic editor with relevant expertise, and I'm writing to let you know that we would like to send your submission out for external peer review.

Once your full submission is complete, your paper will undergo a series of checks in preparation for peer review. After your manuscript has passed the checks it will be sent out for review. To provide the metadata for your submission, please Login to Editorial Manager (https://www.editorialmanager.com/pbiology) within two working days, i.e. by Nov 14 2023 11:59PM.

Kind regards,

Roli Roberts

Roland Roberts, PhD

Senior Editor

PLOS Biology

rroberts@plos.org

---

## [Decision Letter · Decision Letter 1]

21 Dec 2023

Dear Dr Kim,

Thank you for your patience while your manuscript "Single-fly assemblies fill major phylogenomic gaps across the Drosophilidae Tree of Life" was peer-reviewed at PLOS Biology. It has now been evaluated by the PLOS Biology editors, an Academic Editor with relevant expertise, and by three independent reviewers. 

You'll see that all three reviewers are very positive about your study. However, between them they make a number of requests, which you should be able to address relatively quickly. For example, reviewer #1 only has textual requests (provide cost break-down, what about Illumina-only samples, report uncertainties, provide Newick version of tree). Reviewer #2 wants more information on uncertainties all round (SNP/haplotype calls, concordance between seq technologies, concordance with official Dmel assembly) and clearer provision of data. Reviewer #3 wants more discussion of polyphyletic genera, lower-quality sex chromosome assemblies and the ancestral drosophilid.

Based on the reviews, we are likely to accept this manuscript for publication, provided you satisfactorily address the points raised by the reviewers. Please also make sure to address the following data and other policy-related requests.

IMPORTANT:

a) For our wider readership, please include the word "genome" in your title, i.e. "Single-fly genome assemblies fill major phylogenomic gaps across the Drosophilidae Tree of Life"

b) Please address my Data Policy requests below; specifically, we need you to supply the numerical values underlying Figs 1 (treefile), 2, 3, 4, 5, S1 (treefile), either as a supplementary data file or as a permanent DOI’d deposition.

c) Please cite the location of the data clearly in all relevant main and supplementary Figure legends, e.g. “The data underlying this Figure can be found in S1 Data” or “The data underlying this Figure can be found in https://doi.org/10.5281/zenodo.XXXXX”

d) Please make any custom code available, either as a supplementary file or as part of your data deposition.

We expect to receive your revised manuscript within four weeks. 

*Published Peer Review History*

*Press*

Sincerely,

Roli Roberts

Roland Roberts, PhD

Senior Editor,

rroberts@plos.org,

PLOS Biology

DATA POLICY:

Regardless of the method selected, please ensure that you provide the individual numerical values that underlie the summary data displayed in the following figure panels as they are essential for readers to assess your analysis and to reproduce it: Figs 1 (treefile), 2, 3, 4, 5, S1 (treefile). NOTE: the numerical data provided should include all replicates AND the way in which the plotted mean and errors were derived (it should not present only the mean/average values).

CODE POLICY

Per journal policy, as the code that you have generated is important to support the conclusions of your manuscript, we require that you make it available without restrictions upon publication. Please ensure that the code is sufficiently well documented and reusable, and that your Data Statement in the Editorial Manager submission system accurately describes where your code can be found.

DATA NOT SHOWN?

REVIEWERS' COMMENTS:

Reviewer #1:

Review of "Single-fly assemblies fill major phylogenomic gaps across the Drosophilidae Tree of Life"

This was a very interesting paper combining new lab techniques, cutting-edge genomic technologies, and useful genomic data. I really enjoyed reading it, and hope to see it published soon. I had only relatively minor requests for edits or clarifications:

-Please provide a cost breakdown for the $150/sample costs quoted here. I cannot figure out where this comes from. I'm sure the number is low, but I cannot see how it is this low given current ONT flow-cell costs.

-Why does haploid assembly have more base-calling errors than diploid one? Won't a haploid assembly just pick one of the two alleles in a diploid? Doesn't a higher error rate imply it has to pick a third (incorrect) allele?

-Can you say something more about the 16 samples that were only able to be sequenced by Illumina? (Can you also make them easier to identify?) Were these samples stored dry or in ethanol? I could not find a place where the collection status of the tissue for each species was given.

Also, were these samples used in the phylogeny? It is implied (at the bottom of p.14) that they could be used, but it wasn't clear if they actually were.

-I don't see a lot of "uncertainty" in the phylogeny. At worst, there is a branch with a posterior probability of 0.8. Does this branch show evidence for introgression in the Suvorov et al. paper cited? If not, some caution may be warranted with that claim. Can you report the concordance factor for this branch, and all other branches?

-Please provide your phylogeny as a Newick-formatted string in the supplement.

-In the caption of Figure 1 you cite ASTRAL-MP, but in the Methods you cite (but do not name explicitly) ASTRAL-III. Please clarify which was used.

-p. 10 of the pdf (no page numbers were included): the sentence starting "Large section of interesting…" is long and hard to understand. Consider breaking it into two sentences?

Reviewer #2:

This paper highlights development of Oxford Nanopore and Illumina genome sequencing technologies to generate quality genome assemblies from as little as 35 ng of DNA from single flies, and applies the technologies to sequence 183 new genome assemblies for 179 species (of these 121 were from single flies). This is an important advance because of the lack of need to be able to lab culture the flies, so the single fly genomes represent samples direct from natural populations. Data were aggregated with public domain data to generate a phylogeny for 360 drosophilid and 4 outgroup species. They performed a multi-alignment of 298 of these genomes and release it all in an open resource for the research community. 

1. The authors make a convincing argument that diploid assembly performs better than a non-diploid assembly of a pool of flies, even if the sample abundance is limited to the point that the genome is not fully haplotype resolved. But this implies that much of the genomes are resolved as two haplotypes. These regions should be documented/annotated. If a region is called with two haplotypes, then it would be important to know about the SNP calling accuracy. What was the concordance of SNPs from Nanopore vs Illumina reads? The PEPPER-Margin-DeepVariant calls from the Nanopore data could be directly contrasted to the aligned Illumina reads. The downsampling of their 384x melanogaster genome did not get at this. Why not estimate pi from each genome? What was the X vs autosome contrast in nucleotide diversity? Which was better for obtaining the best X chromosome assemblies, single males or single females? Can anything be said about the Y contigs?

2. This reviewer would like to see more detail on the differences between dm6 and their 384x assembly of Dmel. Just giving BUSCO and contig size is pretty limiting. What were the regions of the genome with the biggest discrepancy? Which required the greatest read depth to resolve accurately?

3. How was the phylogenetic tree drawn with the two haplotypes? Were heterozygous sites reported as IUPAC encoding? Or was one haplotype arbitrarily chosen?

4. The authors go to lengths to emphasize that this paper is about a shared resource. This is great, and I applaud the authors for the early release of the data. To maximize the ease of use of the resource, I suggest inclusion of a clear table of data types that are available and their links. Reads, annotated diploid assemblies, the multi-alignment, variant calls, outputs of RepeatModeler2, annotations of haplotype confidence.

5. Full disclosure - this reviewer is far from being an expert in the phylogenetics of Drosophila, so I cannot rate the arguments about taxon sampling or technical details of phylogenetic tree construction at this scale.

Reviewer #3:

The study by Kim et al. represents a significant advancement in genome assemblies for Drosophilidae, providing a valuable resource for researchers in the field. The methodologies employed for sequencing non-culturable species are useful for future research, and the overall clarity of the paper is good. However, I have several suggestions to improve the study before publication further.

First, the presentation of Tables S1 and S4 needs to be improved. Some species only have short read accession numbers like SRR12717852, but no assembly. Can we get the assemblies somewhere? There is also some redundant information between Table S1 and S4. I think the authors can merge these two tables or separate the assemblies from other studies and their studies into two tables. In addition, the authors mentioned that several stocks are contaminated. A dedicated supplementary table listing contaminated stocks and the authors' strategies for resolution would enhance the paper's completeness. If the authors plan to update and correct records, can they provide a link (github?) that allows readers to track? These will be helpful for people who are using the assemblies to notice the possible issues.

Second, as the authors mention, many genera are polyphyletic in Drosophilidae. The authors only stated which species they chose but did not explicitly discuss the alignment of their selected species with previous phylogenetic studies. For example, are there other studies that described four genera as sisters to Drosophila? I understand that the authors might have follow-up studies to talk about introgression or incomplete lineage-sorting, etc., but I think that it is necessary to review some of the previous phylogenic studies and state that their phylogeny is primarily consistent with previous studies here, with some exceptions like D. flavopinicola. Otherwise, judging whether the authors' assemblies can recapitulate what people found before is hard. 

I also noticed that the authors mentioned the less accurate assemblies of sex chromosomes without delving into potential reasons. It might be beneficial if they explore whether the errors stem from differences in coverage between sex chromosomes and autosomes or if the repetitive nature of sex chromosomes plays a role. Offering some insights into the location of errors could be quite enlightening.

Last, the authors mention, "The inferred ancestral drosophilid genome is 33.5 Mbp in size, about 10 Mbp larger than the sum of D. melanogaster coding sequences, and contains 97.3% of the dipteran BUSCO genes as complete and single-copy." I'm curious about the authors' interpretation of this finding.

Minor comments:

1. The authors mentioned: For some of these samples, we made several attempts at a genome assembly and presented the best one here." Can the authors say which these samples are and how they were done? 

2. My Excel can't see the last column of Table S5 

3. The authors have identified X-linked contigs to verify fly sex using Muller elements. It will be great for them to provide the data.

---

## [Editor Report · Decision Letter 2]

3 Jun 2024

Dear Dr Kim,

Thank you for the submission of your revised Methods and Resources "Single-fly genome assemblies fill major phylogenomic gaps across the Drosophilidae Tree of Life" for publication in PLOS Biology. On behalf of my colleagues and the Academic Editor, Chris Jiggins, I'm pleased to say that we can in principle accept your manuscript for publication, provided you address any remaining formatting and reporting issues. These will be detailed in an email you should receive within 2-3 business days from our colleagues in the journal operations team; no action is required from you until then. Please note that we will not be able to formally accept your manuscript and schedule it for publication until you have completed any requested changes.

Sincerely, 

Roli Roberts

Senior Editor

PLOS Biology

rroberts@plos.org